# Magnetically Recoverable Biomass-Derived Carbon-Aerogel Supported ZnO (ZnO/MNC) Composites for the Photodegradation of Methylene Blue

Renathung C. Ngullie [1], K. Bhuvaneswari [2], Paramasivam Shanmugam [1,3,*], Supakorn Boonyuen [3,*], Siwaporn Meejoo Smith [4] and Munusamy Sathishkumar [5]

[1] Department of Chemistry, St. Joseph University, Dimapur 797115, Nagaland, India
[2] Materials and MEMS Laboratory, Department of Electronics and Communication Engineering, Sri Sivasubramaniya Nadar College of Engineering, Kalavakkam 603110, Tamilnadu, India
[3] Department of Chemistry, Faculty of Science and Technology, Thammasat University, Bangkok 12120, Pathumthani, Thailand
[4] Center of Sustainable Energy and Green Materials, and Department of Chemistry, Faculty of Science, Mahidol University, 999 Phuttamonthon 4 Road, Salaya, Nakhon Pathom 73170, Nakhon Pathom, Thailand
[5] Department of Chemistry, Faculty of Science, Chulalongkorn University, Phayathai Road, Bangkok 10330, Pathumwan, Thailand
* Correspondence: shanmugachem@gmail.com (P.S.); chemistrytu@gmail.com (S.B.)

**Abstract:** Hydrothermally assisted magnetic ZnO/Carbon nanocomposites were prepared using the selective biowaste of pomelo orange. Initially, the carbon aerogel (CA) was prepared hydrothermally followed by a freeze-drying method. Furthermore, the iron oxide nanoparticles were deposited onto the surface of carbon using the co-precipitation method and we obtained magnetic carbon nanocomposite, i.e., $Fe_3O_4$/C (MNC). Moreover, the ZnO photocatalysts were incorporated onto the surface of MNC using a hydrothermal process, and we obtained ZnO/MNC composites. The ZnO/MNC (55%), ZnO/MNC (65%) and ZnO/MNC (75%) composites were prepared by a similar experimental method in order to change the weight ratio of ZnO NPs. Using a similar synthetic procedure, the standard ZnO and $Fe_3O_4$ nanoparticles were prepared without the addition of CA. The experimental results were derived from several analytical techniques, such as: X-ray diffraction (XRD), Fourier-transform infrared spectroscopy (FTIR), scanning electron microscopy (SEM), transmission electron microscopy (TEM), Raman and diffuse reflectance spectroscopy (DRS-UV). The synthesized carbon, ZnO, $Fe_3O_4$, ZnO/MNC (55%), ZnO/MNC (65%) and ZnO/MNC (75%) composites were examined through the photocatalytic degradation of methylene blue (MB) under visible-light irradiation (VLI). The obtained results revealed that the composites were more active than carbon, ZnO and $Fe_3O_4$. In particular, the ZnO/MNC (75%) composites showed more activity than the rest of the composites. Furthermore, the recycling abilities of the prepared ZnO/MNC (75%) composites were examined through the degradation of MB under identical conditions and the activity remained constant up to the fifth cycle. The synthetic procedure and practical applications proposed here can be used in chemical industries, biomedical fields and energy applications.

**Keywords:** ZnO/MNC composites; hydrothermal process; organic dyes; biowaste; carbon aerogel

## 1. Introduction

Global industrial growth and modern development create major pollution for the Earth and environment [1]. Water is especially affected by industrial and human-made waste and the consumption of contaminated water causes numerous health issues and death [2]. Contaminants include: organic dyes, pigments, fertilizer, pesticides and heavy metals (Hg, As, As, Cd and Te, etc.) [3,4]. The most extreme pollution occurs from the organic dyes, particularly those used in the textile industry. Many methods have been used for the removal of organic dyes from wastewater. These methods include: the precipitation method,

photocatalysis, solvent extraction, adsorption, ion exchange, nanofiltration, membrane separation, activated carbon filter and reverse osmosis, etc. [5–7]. For the photocatalytic process, various types of materials have been used; in particular, semiconductor materials, such as g-$C_3N_4$, ZnO, ZnS, $TiO_2$, CuO, CuS, CdO and CdS, etc., are mostly used for the removal of organic dyes [8–10]. Particularly, the wide-bandgap semiconductor ZnO NPs (3.37 eV) have the most predominant photocatalytic properties [11]. Additionally, ZnO has been used in solar cells, gas sensors, sunscreens and face creams, etc. [12]. However, the photocorrosion, rapid electron recombination of photogenerated charge carrier, insufficient light absorption, low-efficiency charge separation, recycling and agglomeration are major drawbacks in the photocatalytic process [13,14]. The abovementioned problems can be solved by the addition of supporting materials to the ZnO NPs. Supporting materials, such as nonmetals (C, N, O, B, S and P) [15,16], metals (Ag, Au, Pt, Pd and Ru) [11], metal oxide [17], metal sulfide [18], polymers [19,20] and surfactants, have been used in the previous literature. Particularly, the nonmetals have been shown to enhance the activity and electron recombination rate [21,22]. Biomass-derived carbon and carbon aerogels have received significant attention in photocatalytic degradation [23,24]. Carbon aerogels (CAs) have a highly porous structure, are light-weight, are non-toxic, require a simple preparation procedure and have 3D carbon canulate networks [15,25]. The 3D carbon network helps to enhance the electrical, mechanical and thermal properties [26]. Additionally, the biowaste-derived carbon aerogel contains an abundant number of functional groups such as -OH, -COOH, -C=O, -$NH_2$ and C-O-C, which enhance the stability of the nanoparticles and reduce the particle size [27,28]. The carbon materials are prepared by various techniques such as chemical vapor pressure, laser ablation, arch-discharge, pyrolysis, combustion and the hydrothermal process [27,29,30]. Among them, the hydrothermal process followed by the freeze-drying process is the predominant method for the preparation of CA due to it being non-toxic, readily available and economically feasible [27,30,31]. Recently, Yuning et al. explored CA/$TiO_2$ as an efficient photocatalyst for the removal of organic dyes [32]. Furthermore, in increasing the recycling process, magnetic nanoparticles can be attached to biomass-derived carbon [33]. This paper reveals that magnetic particles can incorporate biomass-derived carbon-aerogel-supported ZnO (ZnO/MNC) composites through extreme hydrothermal processes, with the biowaste of pomelo orange used as a carbon source.

## 2. Experimental Section

### 2.1. Materials Required

Fresh ponderosa lemon was purchased from Dimapur local supermarket, Nagaland, India, and washed with DD water before use. Ferric Chloride ($FeCl_3 \cdot 6H_2O$), Ferrous Sulphate ($FeSO_4 \cdot 7H_2O$), Zinc sulphate ($ZnSO_4$), ammonia solution, ethanol and sodium hydroxide (NaOH) were obtained from SRL chemical PVT. Ltd., Mumbai, India. The deionized (DI) water was used for all reactions.

### 2.2. Preparation Magnetic ZnO/MNC Composites

The synthesis of carbon aerogel (CA) and $Fe_3O_4$/carbon (MNC) composites were prepared by using prescribed synthetic procedure [20,23]. The ZnO/MNC composites were prepared by simple hydrothermal process [5] (Scheme 1). The 2.68 g of $ZnSO_4 \cdot 7H_2O$ and 4 g of NaOH were dissolved in 100 mL of water. Then, the NaOH solution was added into the $ZnSO_4 \cdot 7H_2O$ solution drop by drop, the whole reaction mixture was stirred magnetically, and the homogeneous solution was converted into heterogeneous white precipitate. Then, the required amount of MNC was added into the reaction mixture. The whole reaction was moved to the Teflon-lined stainless-steel autoclave and kept at 180 °C for 12 h. After cooling it to room temperature, the obtained ZnO/MNC hydrogel was washed with a $C_2H_5OH$ and $H_2O$ mixture (1:1 ratio) to remove the soluble impurities. The ZnO/MNC composites were obtained by the high-vacuum freeze-drying method at −52 °C for 36 h. A similar method was used to prepare different composites to change the concentration

of $ZnSO_4$, thus obtaining 55%, 65% and 75% ZnO/MNC composites. Furthermore, the pure ZnO nanoparticles were prepared without the addition of MNC, using a similar synthetic procedure.

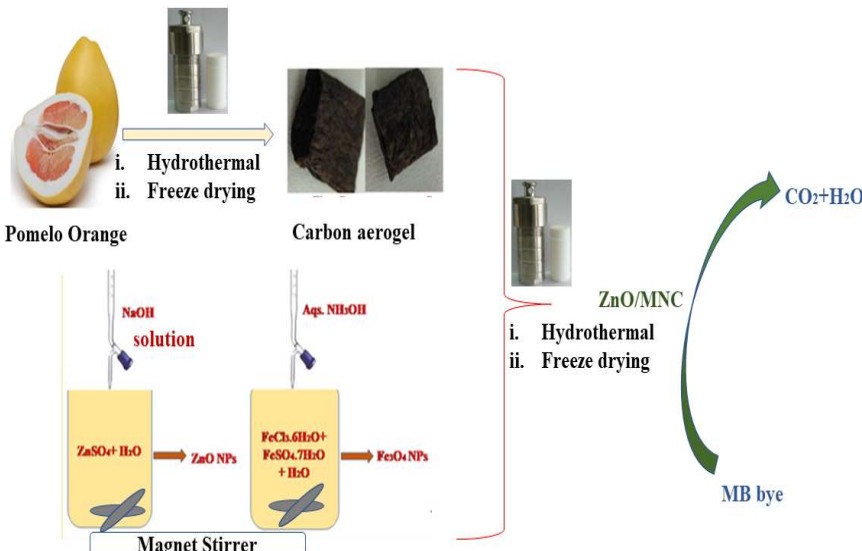

**Scheme 1.** Synthesis and photocatalytic applications of ZnO/MNC composites.

### 2.3. Photocatalytic Study

The MB was chosen as the model organic pollutant to examine the photocatalytic efficiency of carbon, ZnO, $Fe_3O_4$, ZnO/MNC (55%), ZnO/MNC (65%) and ZnO/MNC (75%) composites. The UV light irradiation was applied by 500 w Xe lamb. To control the photocatalytic activity, 10 mg of ZnO/MNC (75%) was immersed into 100 mL of the MB (10 mg/L) solution. The reaction mixture was stirred for 1 h in dark conditions to reach ad/desorption equilibrium. The dye solution was then treated with UV-visible-light irradiation. After that, 3 mL of suspension was taken at time intervals of 30 min; the suspension was taken and centrifuged to eliminate the photocatalyst before being stored in the freezer. The evolution of the photodegradation was observed by UV-visible spectrophotometry and by measuring the decrease in the absorbance intensity at 662 nm. Similar photocatalytic studies were carried out using the rest of the materials to find the suitable photocatalyst, based on the obtained results. The photodegradation of MB dye was carried out by $C/C_0$; where C and $C_0$ are the maximum absorption of MB at the initial time and at the after-irritation time (t), respectively.

### 2.4. Characterization

The X-ray diffraction (XRD) study was performed using Rigaku ULTIMA 1V IR-Technology service Pvt. Ltd. FT-IR spectra were recorded by FTIR on a Cary 630 FTIR Agilent Technologies India Pvt. Ltd. The field emission scanning electroscope (FESEM) was from Carl Zeiss, Sigma, Germany. High-resolution transmission electron microscopy (HR-TEM) was performed on a JEOL: JEM2100 PLUS instrument, Tokyo, Japan, operating at 200 kV. Raman spectrometer equipped with a He-Ne laser with an irritation of 532 nm was used for attaining the Raman spectra (Thermo Fischer DXR, USA). A SHIMADZU 3600 UV-visible spectrophotometer analyzed the energy bandgap and absorption of the prepared composites. The emission profile of the prepared composites as obtained with the help of a photoluminescence (PL) investigation using Perkin Elmer-LS45 spectrometer.

## 3. Results and Discussion

### 3.1. Characterization of ZnO/MNC Composite Photocatalysts

The biowaste-derived CA was prepared using a simple hydrothermal process (Scheme 1). During the hydrothermal process, the cross-linked poly-saccharose, polypeptides and

polyphenolic compounds in the soft tissue of pomelo oranges were converted into carbon hydrogel [14]. Shanmugam et al. reported the use of a biomass-derived magnetic carbonaceous aerogel in the removal of organic pollutant [25]. Furthermore, the water molecules were removed by the freeze-drying process and the hydrogel of carbon was converted into carbon aerogel. The obtained CA was used as an excellent scaffold/supporting material for the preparation of ZnO/MNC composites. The XRD patterns of the prepared C, ZnO, $Fe_3O_4$ and ZnO/MNC composites are given in Figure 1a. The main peaks observed at 31.61°, 34.5°, 36.23°, 47.5°, 56.45°, 62.86° and 68.01° can be observed in ZnO, and were indexed as (100), (002), (101), (102), (110), (103) and (201) planes of hexagonal wurtzite structure (JCPDS 36-1451) [5]. Furthermore, the characteristic peaks of $Fe_3O_4$ which appeared at 30.2°, 35.6°, 43.3°, 57.1° and 62.8° were observed to the (220), (311), (400), (511) and (400) lattice planes of the magnetite spinel structure of $Fe_3O_4$, fixed with the JCPDS value of 19-0629 [5,34]. In Figure 1a, the broad characteristic peak appeared at 22.8°, which is the structure of amorphous carbon, and is similar to graphitic carbon [15,35]. After the combination of ZnO and MNC, all the ZnO/MNC samples displayed diffraction peaks of both carbons and ZnO on the MNC surface [25,34]. Samaneh et al. reported similar results in the graphene oxide/$Fe_3O_4$/ZnO ternary composites for the photodegradation of metalaxyl. The crystallinity and intensity of ZnO increased with the increase in the ZnO metal loaded onto the MNC. Additionally, the results of the ZnO/MNC composites revealed that the crystallinity of CA did not alter the ZnO crystallinity. Furthermore, no additional peaks appeared in the ZnO/MNC composites; thus, the results evidenced that the prepared ZnO/MNC composites did not contain any impurities. By using the Scherrer equation, the average crystallite was calculated: 43 nm for ZnO and 26 nm for ZnO/MNC, respectively. Moreover, on comparing the size of ZnO and ZnO/MNC composites, the size of the ZnO nanoparticles was smaller in ZnO/MNC composites than pure ZnO, because the biomass-derived carbon aerogel (CA) reduced/stabilized the size of the ZnO nanoparticles.

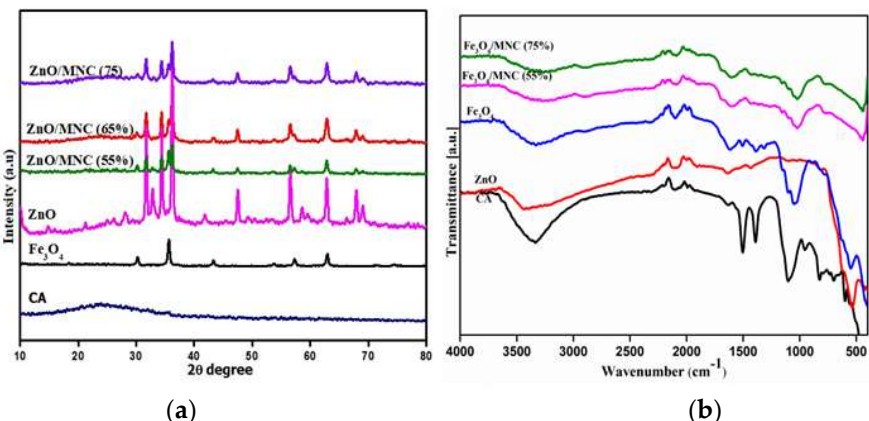

(a)   (b)

**Figure 1.** (**a**) XRD pattern; (**b**) FTIR spectra of pure carbon, $Fe_3O_4$, ZnO, ZnO/MNC (55%), ZnO/MNC (65%) and ZnO/MNC (75%) composites.

The surface-functional group of the CA, ZnO, $Fe_3O_4$ and ZnO/MNC (55%), ZnO/MNC (65%) and ZnO/MNC (75%) composites were analyzed by FTIR analysis in the range of 4000–400 $cm^{-1}$, which is illustrated in Figure 1b. It was employed to specify the functional group of the CA. The IR peaks of the solid samples are related to the vibration of ions in the photocatalytic structure. The two peaks at 535.48 $cm^{-1}$ and 548.37 $cm^{-1}$ are associated with the ZnO and Fe-O stretching vibration [36,37] (Figure 1b). The characteristic peaks appear at 1023, 1154, 1315, 1398, 1506, 1618 and 3338 $cm^{-1}$ due to -OH functional group, aliphatic ether, C-O, aromatic C=C, carbonyl C=O and –OH hydroxyl groups, respectively. These results support that the CA containing -OH, C=O, R-O-R and aromatic groups, which are used as scaffold materials for the functionalization of nanomaterials, improved the hydrophobic and dispersible properties in water [14]. The two kinds of metal-oxide-absorption peaks were identified, approving the development of Zn-O and Fe-O (Figure 1b). These Zn-O

and Fe-O absorption peaks appeared with all of the ZnO/MNC composites, thus confirming that the ZnO and $Fe_3O_4$ NPs are strongly incorporated onto the surface of carbon (Figure 1b). Furthermore, the -OH and –C=O values slightly shifted from 3338 to 3342 $cm^{-1}$ and from 1618 to 1613 $cm^{-1}$, thus ZnO and $Fe_3O_4$ may be incorporated/stabilized into the –OH and C=O functional group [34]. Additionally, to increase the % composition of ZnO, the intensity of -OH and –C=O was decreased, and Zn-O and Fe-O were increased.

The surface morphology of the prepared carbon, ZnO, $Fe_3O_4$ and ZnO/MNC (75%) composites are shown in Figure 2a–d. The SEM images of the CA show a 3D carbon network, a highly porous structure, globular canulate and a smooth surface at the 0.55 µm size. The highly meso- and micro-porous carbon aerogel structure enhanced the adsorption and active-metal-loading properties [14,25]. Furthermore, the $Fe_3O_4$ nanoparticles were incorporated onto the surface of CA and thus $Fe_3O_4$/CA was obtained. The SEM images of the $Fe_3O_4$/CA composites reveal that a large number of block spheres appeared on the carbon nanofiber, thus confirming that the $Fe_3O_4$ was strongly attached to the nanofiber of CA, due to the large number of functional groups present in the surface of the carbon. Furthermore, the ZnO was encapsulated/incorporated onto the surface of MNC and thus ZnO/MNC was obtained; the corresponding images are shown in Figure 2c,d. According to the SEM images of the ZnO/MNC composites, the smooth surface of the carbon aerogel was converted into a heterogeneous surface due to the incorporation of ZnO and $Fe_3O_4$. Further, black and white dots appeared on the carbon surface. The average particle size was approximately 25–30 nm. On comparing the pure ZnO and its composites, the particles sizes decrease due to the stabilization of the carbon aerogel. Furthermore, the SEM images of ZnO and $Fe_3O_4$ revealed that the maximum number of ZnO particles were strongly attached to the carbon nanofiber, both micro- and mesoporous, of the carbon aerogel.

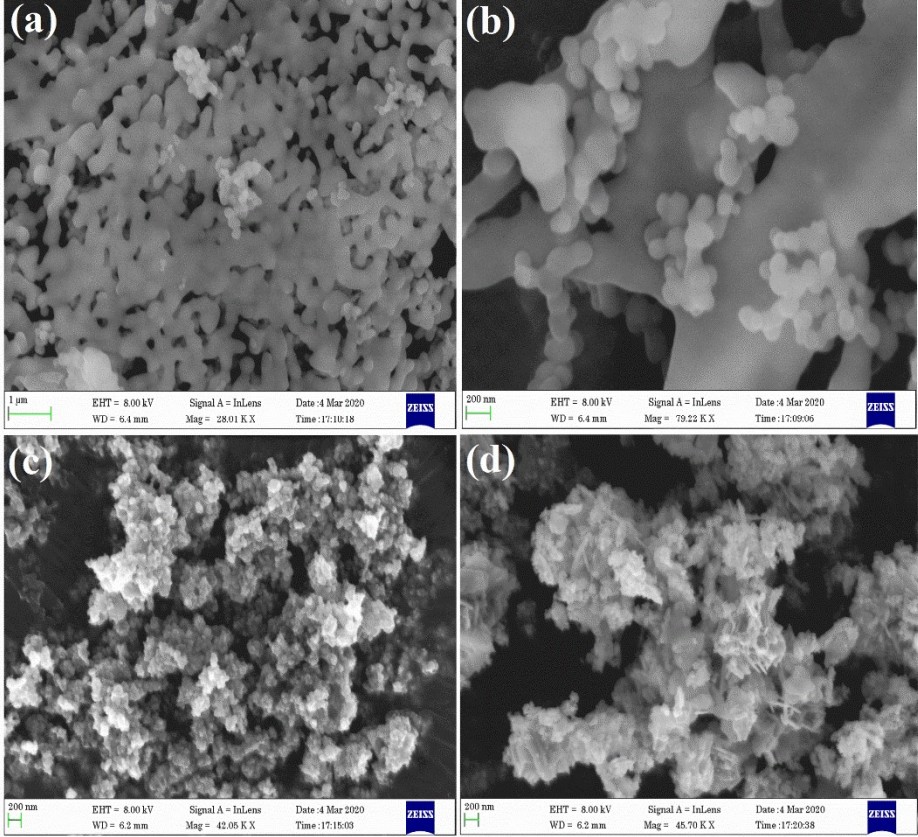

**Figure 2.** FE-SEM images of: (**a**,**b**) CA and (**c**,**d**) ZnO/MNC (75%) composite materials.

Furthermore, the size and shape of the ZnO/MNC composites were analyzed by the HRTEM study and the images are shown in Figure 3a–e. The low-magnification HRTEM

images of the ZnO/MNC are shown in Figure 3a,b. The images show that the particles were spherical in shape, with the ZnO and $Fe_3O_4$ nanoparticles strongly bound to the surface of the CA. Furthermore, all the nanoparticles were evidently distributed on the surface of the carbon network [38]. The obtained particle size of the ZnO was approximately 28 nm. The HRTEM results accurately matched the XRD and SEM analysis results. Figure 3e shows the selected diffraction pattern of the ZnO/MNC composites. Thus, the images revealed that the ZnO/MNC composites materials were crystalline in nature. Particularly, the bright spot shows the crystalline nature of the ZnO nanoparticles. Additionally, some ring patterns were observed in the SAED pattern, which can be attributed to the presence of carbon.

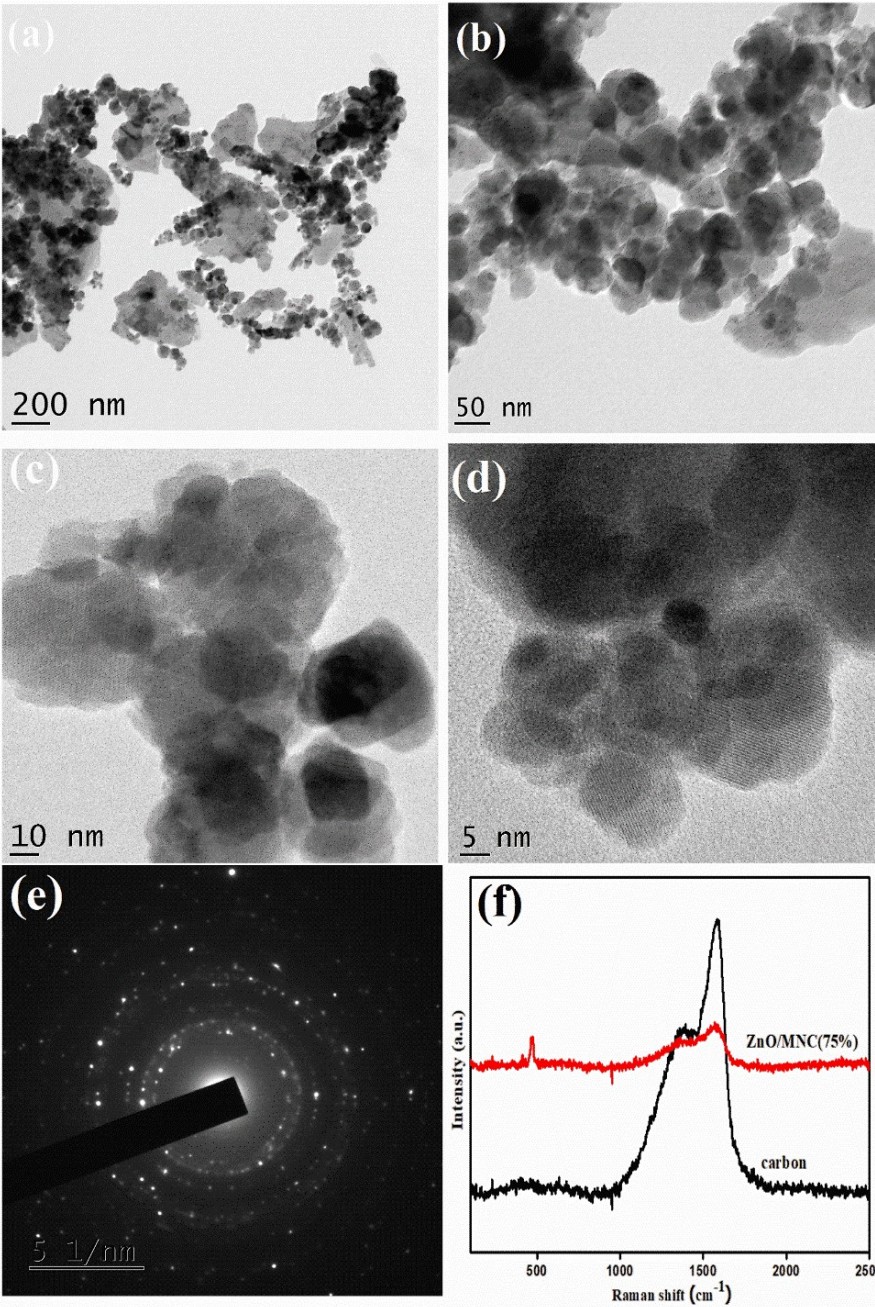

**Figure 3.** HR-TEM images of: (**a**,**b**) ZnO/MNC (75%) (low magnification), (**c**,**d**) ZnO/MNC (75%) (high magnification) and (**e**) SAED diffraction pattern of ZnO/MNC (75%); (**f**) Raman analysis of the carbon and ZnO/MNC (75%) composites.

The Raman spectra of the carbon and ZnO/MNC composites showed characteristic peaks for the carbon (Figure 3f). The characteristic peaks observed at 1376 cm$^{-1}$ (D band) and 1784 cm$^{-1}$ (G Band) were associated with structural defects/disorders of sp$^3$ carbon hybridization and the planner sp$^2$ carbon structure, respectively. The intense peak observed at 468 cm$^{-1}$ was assigned to the non-polar optical phonon E2(HI) vibrational mode of zinc oxide in wurtzite structure [39]. Furthermore, the ZnO was strongly bound to the surface of MNC (Figure 3f). The intensity of the Raman spectra of carbon was affected by the inchoation of ZnO and $Fe_3O_4$, and gradually decreased with the increase in the ZnO and $Fe_3O_4$ contents, which was also reported in [40]. The $I_D/I_G$ values were calculated using intensity of the D and G band values. The obtained $I_D/I_G$ values for carbon and ZnO/MNC were 0.96 and 0.91, respectively. With the incorporation of ZnO nanoparticles onto the MNC matrix, the $I_D/I_G$ values is reduced from 0.96 to 0.91, which is due to sp$^2$ carbon structural defects occurring during the hydrothermal process.

The intense peak that appeared at 440 cm$^{-1}$ was assigned to the non-polar optical phonon E2 (HI) vibration mode of ZnO in the wurtzite structure.

### 3.2. Photophysical Properties

The photo-absorption capacity and bandgap of the prepared samples were studied through UV-visible analysis, and their observed results are shown in Figure 4. In the original ZnO nanoparticles, the absorption maximum was observed in a range from 200 to 380 nm, which confirms the absorption characteristics of the ZnO nanoparticles. In Figure 4, the absorption wavelength of $Fe_2O_3$ shows higher absorption in the visible-light region compared to pristine ZnO nanoparticles. Similarly, pure carbon nanoparticles showed moderate absorption in the UV region as well as in the visible-light region. Composites ZnO/MNC (55%), ZnO/MNC (65%) and ZnO/MNC (75%) showed improved absorption of visible light compared to the pristine and composite samples. Notably, the ZnO/MNC (75%) nanocomposites (Figure 4a) showed increased absorption in the visible light. The observed results showed that the ZnO/MNC nanocomposites (75%) had a high generation of photogenerated charge carriers under the influence of visible-light absorption. Compared with the pristine ZnO, the absorption edge of the prepared composites increased from 400 to 800 nm, which indicates a redshift in absorption. This observed redshift was due to the modification of the electronic state in the ZnO crystal structure due to the presence of a carbon atom, which can relatively improve the visible-light absorption results. The energy bandgap of the prepared samples was calculated by Tack's plot method and the calculated bandgaps of the prepared samples were 3.17, 2.77, 2.86, 2.71, 2.6 and 2.5 eV for the ZnO, $Fe_3O_4$, carbon, ZnO/MNC (55%), ZnO/MNC (65%) and ZnO/MNC (75%) composites, respectively (Figure 4b). Hence, the observed results conclude that ZnO/MNC (75%) composites can modify the electronic state and boost the visible-light absorption ability with its subatomic energy band positions of ZnO, $Fe_3O_3$ and carbon nanoparticles.

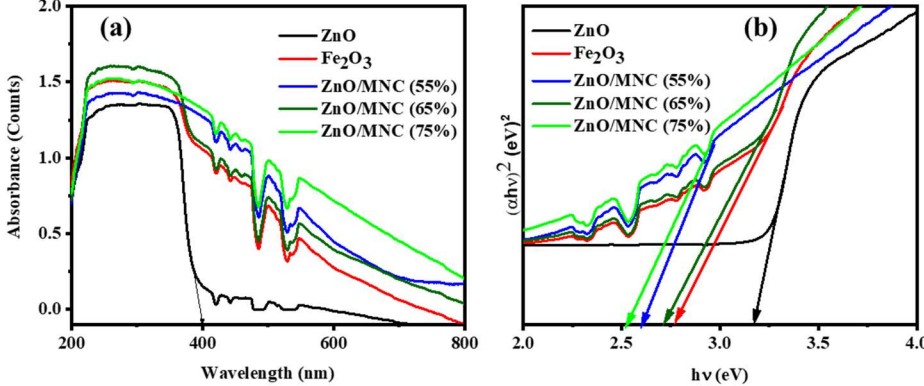

**Figure 4.** (**a**) UV-DRS and (**b**) Tauc plot for prepared samples.

In the photocatalytic degradation process, the charge carrier recombination rate influences the degradation efficiency of the prepared samples. Hence, the charge carrier recombination profile of the prepared samples was investigated by photoluminescence (PL) analysis with an excitation wavelength of 360 nm. The outcomes were shown in Figure 5. The intense profile of the prepared samples denoted the high charge carrier recombination rate of the sample, whereas the low intensity reveals the slow electron–hole recombination rate. Figure 5 shows the PL spectrum of the pure ZnO, ZnO/MNC (55%), ZnO/MNC (65%) and ZnO/MNC (75%) composites. The emission band located at 388 nm caused the charge carrier recombination at ZnO nanoparticle. The pure ZnO nanoparticles and ZnO/MNC (55%) and ZnO/MNC (65%) showed higher luminescence intensity which can be attributed to fast charge carriers' recombination, compared to the ZnO/MNC (75%) composite. This outcome suggests that the 65 (%) of MNC helped to reduce the electron–hole recombination of ZnO, which is advantageous in enhancing the degradation performance.

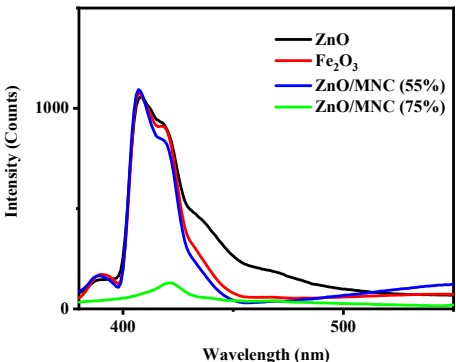

**Figure 5.** Photoluminescence spectra for prepared samples.

### 3.3. Photocatalytic Study

The kinetics of the MB dyes determined by the prepared catalyst, the obtained results were fitted by pseudo-first-order model, $\ln(C_0/C) = kt$, where $C_0$ and $C$ represents the initial and final concentration(t) of MB, and k represents the first-order rate constant (Figure 7). The reversion of $\ln(C_0/C)$ against time is linear, and its slope equals the first-order apparent rate constant (K). Figure 6 reveals the difference in the percentage of photodegradation efficiency with illumination time for the photodegradation of MB. The as-prepared ZnO/MNC (75%) sample exhibited a higher photocatalytic degradation efficiency (97.14%) against MB than the pure CA (13.27%), ZnO (43.28%), $Fe_3O_4$ (18.44%), ZnO/MNC 55% (70.8%), and ZnO/MNC 65% (80.57%) samples under the same conditions. The values of maximum degradation efficiency of MB are listed in Table 1 where previous reports are also included for better understanding.

### 3.4. Recycling Efficiency of the ZnO/MNC (75%)

The stability and activity of the prepared materials were checked using a recycling study. Reusability is an important parameter for ensuring effective environmental characteristics for practical utilization. After the first cycle, the photocatalysts were collected using external magnetic field and washed, dried, and recycled. Figure 7c displays five sequential recycling tests for photocatalytic degradation of MB over ZnO/MNC (75%). The mild changes occurred in the recycling activity because of defusing, which may be attributed to the loss of materials in the centrifugation process. In comparison with the first cycle (97.14%), the photocatalytic degradation efficiency of ZnO/MNC (75%) still exceeded 91% after five consecutive cycles. Furthermore, Figure 7d reveals that the magnetic property of the ZnO/MNC (75%) composites remained constant up to the fifth cycle. The magnetically recoverable composite materials can be used several times for the degradation of organic dyes and industrial effluent treatment.



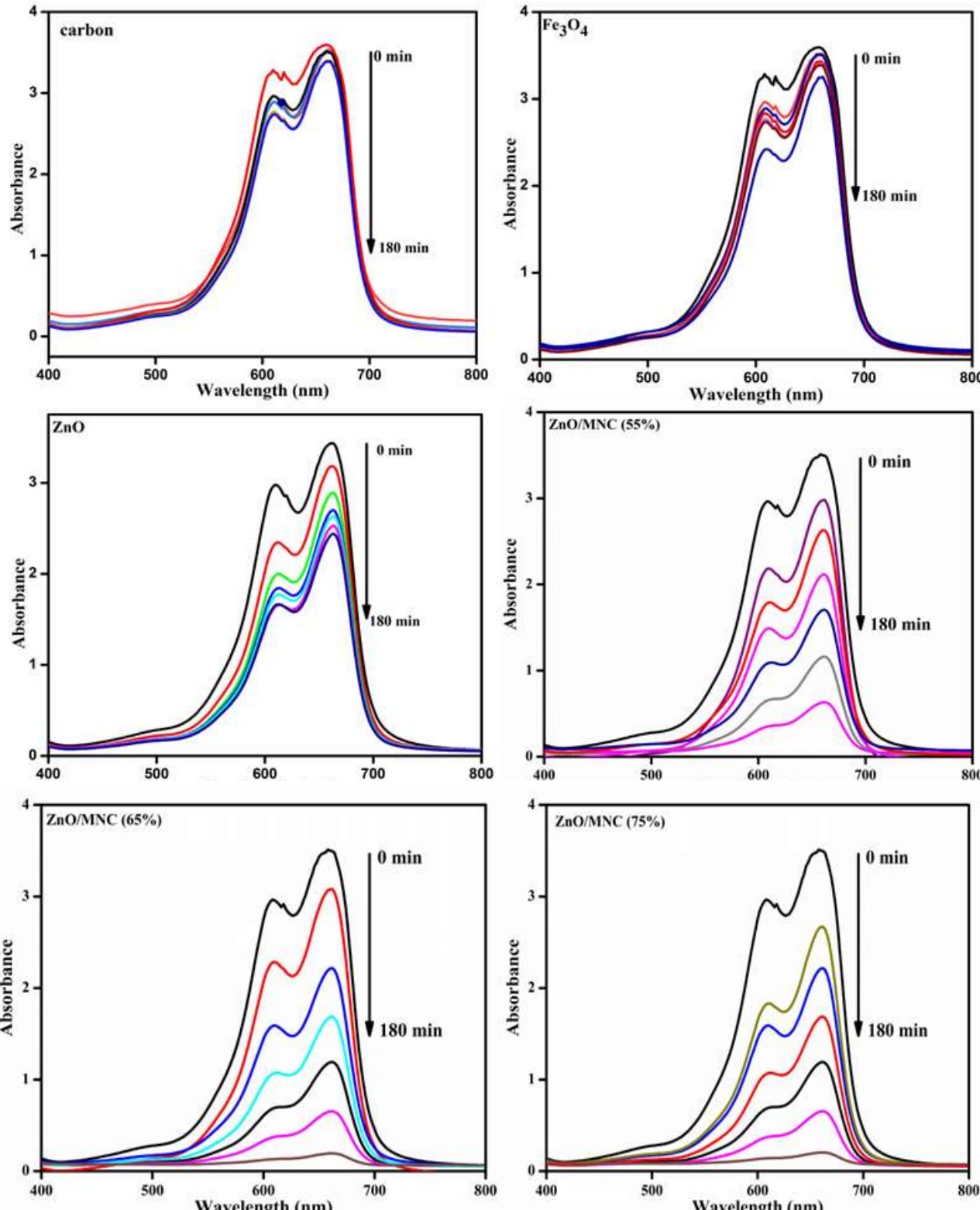

**Figure 6.** UV-visible absorbance spectra of visible-light active photodegradation of MB dye in the presence of as-prepared samples.

**Table 1.** Comparison of MB photodegradation of different photocatalysts.

| S. NO | Photocatalysts Name | Percentage of Efficiency | Reference |
|---|---|---|---|
| 1 | $ZnO/Fe_3O_4$ | 63.02 | [41] |
| 2 | $ZnO/Fe_3O_4$ | 88.5 | [42] |
| 3 | $ZnO/GO$ | 88 | [43] |
| 4 | $ZnO/g\text{-}C_3N_4$ | 98 | [5] |
| 5 | $Pt\text{-}ZnO/MWCNT$ | 74 | [44] |
|  | $ZnO/CS$ | 80.34 | [45] |
| 6 | $ZnO/MNC$ | 97.14 | This work |

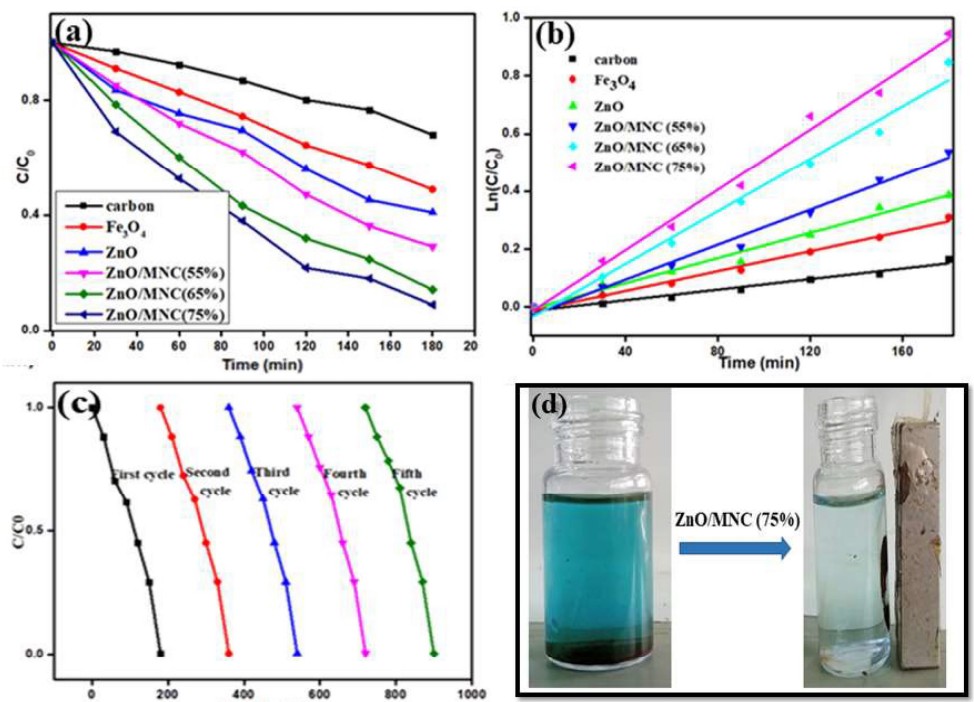

**Figure 7.** (**a**). Photodegradation of MB dye; (**b**) pseudo-first-order kinetic plots over the presence of the prepared samples; (**c**) studies on the recycling stability of ZnO/MNC (75%) composites over MB dye; (**d**) magnetic recoverability of the photocatalyst using external magnetic field.

Furthermore, compared with $Fe_3O_4$, $F_3O_4/CA$ and ZnO/MNC composites, the ZnO and ZnO/CA cannot be separated magnetic fields. Centrifugation or filtration is necessary to separate non-magnetic photocatalysts. Besides being time-consuming, mass loss is another disadvantage of centrifugation and filtration in the recycling process. Using centrifugation/filtration process and the magnetic separation process, the recovery of the catalyst is 70% and 95%, respectively. This amounts to almost 25% higher catalyst recovery for the magnetic photocatalysts. All of these outcomes have revealed that the high catalytic activity of $ZnO/Fe_3O_4/CA$ is derived from the absorption of photons by semiconductors, the rapid migration of electron–holes by $Fe_3O_4$ and carbon, and enhanced adsorption of target molecules by ZnO and CA. Sonal et al. achieved the 85.7% photodegradation efficiency using ZnO/AC composites [46]. Compared with previous literature, we achieved 97.14% degradation efficiency using magnetically recoverable ZnO/MNC composites, due to magnetic $Fe_3O_4$ coupled with ZnO to reduce the bandgap, which enhanced the photon-adsorption efficiency and potentially increased the surface area of the photocatalysts. Therefore, the ZnO/MNC composites performed as a robust, recyclable and effective catalyst for photodegradation.

### 3.5. Mechanism

The photocatalytic-dye-degradation mechanism of MB over the prepared ZnO/MNC composites is shown in Figure 8. When $Fe_3O_4$ is coupled with ZnO NPs, the proton ($h^+$) may be absorbed in both $Fe_3O_4$ and ZnO to form the electron–hole pairs. The electrons of the conduction band (CB) of $Fe_3O_4$ could migrate to that of ZnO. However, the $h^+$ of the valance band (VB) of $Fe_3O_4$ remains at the same position. This process reduces the probability of the charge carrier recombination. At the same time, the free electrons ($e^-$) were transferred to the carbon aerogel, which act as an electron collector as well as donor. This reduces the electron–hole pair recombination rate, thus enhancing the photocatalytic efficiency of the ZnO/MNC composites when compared with pure ZnO. Furthermore, the hydroxyl ($OH^\bullet$) radical present in the aqueous solution is captured by the $h^+$ of the VB to form highly reactive hydroxyl radical. The toxic dye molecules are then targeted by this radical, then converted into non-toxic simple products.

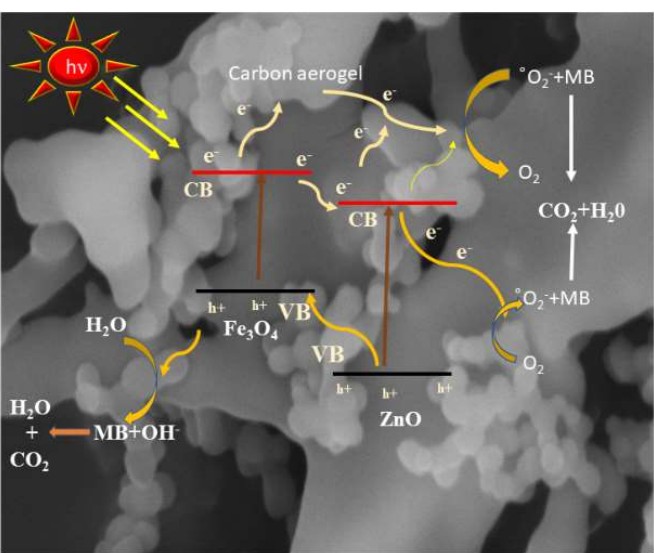

**Figure 8.** Photodegradation mechanism of methylene blue under UV-visible light by ZnO/MNC composites.

## 4. Conclusions

In summary, the ZnO/MNC composites were successfully prepared by a hydrothermal process in which ZnO nanoparticles embedded onto the surface of CA. The ZnO/MNC composites materials showed better photocatalytic activity than pristine materials. In particular, the ZnO/MNC (75%) composites activity increased twofold compared with pure ZnO and carbon, due to the composite materials containing a strong visible-light absorption profile, a redshift light-absorption range, the possibility for enhancing the electron–hole pair separation rate, as oriented from the suitably similar conduction band, and the valence band potential between ZnO and carbon. Additionally, the enhanced light absorption with an appropriate energy band position could enhance the photocatalytic activity of the prepared samples. Similarly, 65 (%) content of MNC helped to reduce the electron–hole recombination rate of the photocatalyst, which can improve the interaction between the photogenerated electrons and the dye molecules. Furthermore, the magnetic $Fe_3O_4$ nanoparticles also enhance the activity of ZnO. Through completion of the photocatalytic process, the photocatalysts were recoverable through an external magnetic field. In addition to demonstrating potential for reusability and stability, the prepared ZnO/MNC composites also showed high stability.

**Author Contributions:** R.C.N.: data collection, writing—original draft preparation, methodology and investigation; K.B.: resources, writing—review and editing; P.S.: data curation, writing—original draft preparation, methodology and investigation; S.B.: methodology, conceptualization, supervision and writing—review; S.M.S.: review and resources; M.S.: writing—review and editing. All authors have read and agreed to the published version of the manuscript.

**Funding:** This study was supported by Thammasat Postdoctorla Fellowship (Grand No. B.E.2564) and Thammasat University Research Fund, Contact No: TUFT-FF 4/2565.

**Data Availability Statement:** Not applicable.

**Acknowledgments:** The authors gratefully acknowledge the Thammasat University Postdoctoral Fellowship (B.E.2564) and Thammasat University Research Fund (TUFF-FF 4/2565). Further, the authors thank to NIT Nagaland and St. Joseph University to supports continuous characterization techniques.

**Conflicts of Interest:** The authors declare no conflict of interest.

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
