# Peer review of "Magnetically Recoverable Biomass-Derived Carbon-Aerogel Supported ZnO (ZnO/MNC) Composites for the Photodegradation of Methylene Blue"

_catalysts, doi:10.3390/catal12091073_

Round 1
Reviewer 1 Report
The authors prepared a biomass carbon aerogel supported ZnO and iron oxide as the photocatalyst to degradation dye MB. This work is interesting, the characterizations were abundant, and the result was obvious. I suggest it could be accepted after some minor revisions.
1. Abstract, the first thee sentences introduced the preparation of the composite, which were verbose, please revise these sentences.
2. The English language and grammar should be carefully revised.
3. Abstract, the full names of characterizations should be given, rather than only the abbreviations.
4. Introduction, the authors should supplement some research development of aerogel.
5. Figure 6, every figure should be labeled using its catalyst name.
6. The photocatalytic mechanism about the reactive species should be discussed, or supplement some experiments.
Author Response
Response to the Reviewers’ comments
Reviewer #1:
Comment 1: Abstract, the first three sentences introduced the preparation of the composite, which were verbose, please revise these sentences.
Response: Thanks for your valuable comments, as per your suggestion the first sentence of the abstract was revised in the manuscript
Comment 2: The English language and grammar should be carefully revised.
Response: Thank you for your valuable comments, the English language and grammar were thoroughly checked by native English speakers.
Comment 3: Abstract, the full names of characterizations should be given, rather than only the abbreviations.
Response: Yes, we agree with our mistake, as per your suggestion we gave the full name of the characterization in the revised manuscript.
Comment 4: Introduction, the authors should supplement some research development on aerogel.
Response: Thanks for your comments, as per your suggestion, we were given more information about the carbon aerogel in the revised manuscript.
Comment 5: Figure 6, every figure should be labeled using its catalyst name.
Response: We have changed it in the revised manuscript.
Comment 6: The photocatalytic mechanism about the reactive species should be discussed, or supplement some experiments.
Response: These are valuable comments to improve the quality of the manuscript. We proposed a photocatalytic mechanism and discussion of the prepared compared ZnO/MNC composites in the revised manuscript.
Reviewer 2 Report
This manuscript is reporting on the preparation and photochemical reactions of ZnO supported by magnetite/carbon aerogel composites. R. C. Ngullie et al. investigated the conventional analysis of prepared materials, and the photocatalytic ability of the obtained materials was confirmed using methylene blue as a standard dye (pollutant) molecule under visible light (l > 420 nm) illumination. ZnO is known to have several drawbacks, but the current method seems to overcome the problems to show better photocatalytic performance over control systems. In addition, its “magnetic” recovery was also demonstrated. Thus, this work will be a good piece towards developing efficient photocatalysts (although lacking the magnetic and surface area analysis). I recommend the publication of the manuscript in Catalysts after revision. A few comments are listed below:
Major points:
1) “Magnetically” is not well supported: First, in line 242 the recycling using magnetic field is discussed. What’s happened without “magnetic field”? Discuss more details for this process, because this would be the most important part in this work. Second, in line 65, the authors mentioned to the magnetic properties in relation to the recyclability. Indeed, the photocatalytic durability was supported by data shown in Figure 7c. However, if the authors assert the magnetic property-related recyclability, it would be better for the authors to show the magnetic property of materials including saturation magnetization (ss) before and after the photochemical investigation. For example, ss of bulk magnetite is reported to be 90 emu g‒1 (Angew. Chem. Int. Ed. 2007, 46, 1222–1244), and its comparison and detailed experimental description would greatly support the “magnetic recyclability.”
2) in line 173, the authors discussed the “improved absorption of visible light compared to the pristine.” However, this seems to be just due to the additional absorption by Fe3O4 (and “carbon”). Analysis for Figure 4b is also queer. In addition, ZnO itself also showed a certain photocatalytic performance as shown in Figure 6c. If the authors discuss the “seemingly narrowed” optical bandgap in relation to the improved photocatalysis, the authors must conduct photochemical investigations with, for example, l > 450 nm using a cut-filter. Otherwise, the whole discussion of the energy bandgap using Tack’s plot doesn’t make sense.
3) In line 17, the authors mentioned “nanocomposites were designed and synthesized.” However, it was not clear for what is the design philosophy for this work (in other word, what is designed by how based on which idea?).
4) The authors first mentioned to XRD and IR of materials, and then discussed the preparation. This sequence is less friendly for readers to understand the contents. Please first mention to how to prepare C, ZnO, and ZnO/MNC in the Results and Discussion part, with appropriate citations, if any.
5) “Advantage of carbon” is less clear in the introduction part: In line 63, the authors mentioned to “biomass derived carbon” and carbon aerogel for previous works, but the advantage and rational necessitation of using them is unclear to readers. And, what’s the advantage of this method over many other synthetic methods, including chemical vapor deposition (Chem. Sci. 2022, 13, 3140‒3146; Adv. Mater. 2022, 34, 2108750)? If the current method enjoys relatively cheap cost with relatively large surface area and enriched polar functional groups including hydroxyl and carbonyl groups as is discussed in lines 96-106, then they would also be great advantages of utilizing them in this work, and this should be mentioned in the introduction part for a “better reading material”.
6) In lines 74 & 77, there are description of Miller indices such as (100), (002), and (101). The parenthesis means a specific plane in definition. If the XRD patterns were obtained for isotropic powder, then the correct description will be 100, 002, and 101 rather than (100), (002), and (101). For details, please refer to a suitable text book. This would be the same for the selected area electron diffraction in Figure 3e, if further characterization is accomplished for the SAED.
Minor points:
1) In Figure 3e, there are many spots in the selected area electron diffraction. Because the current work lacks EDS analysis for SEM/TEM, this SAED is a very nice source of identifying and characterizing the composites at nano-level. The referee recommends the authors to analyse the SAED for better characterization, if possible.
2) In figure 5, what’s “Caron)”? If this is correctly carbon, then why do the carbon and magnetite show the photoluminescence similar to that of ZnO in the steady-state emission spectra? Maybe they are correctly Zn/MNC(##%), then unify the color in figures throughout the manuscript (For example, using the same colors for each material in Figure 4 and Figure 5 will help authors understand well the content).
3) in line 55, the authors described “rapid electron recombination of photogenerated charge carrier,” but this may correctly be “rapid charge recombination of photogenerated hole-electron pair,” for example.
4) in line 64 and more several times later, there are a word “CA,” but it was not defined throughout the manuscript.
5) in lines 23 & 91 (and others), there is an unclear word like “ZnO/MNC(55%).” What does the percentage value mean? Clearly define and explain them at both the introduction and main text (otherwise readers cannot understand what the sentence means).
6) A Raman spectrum in Figure 3 is denoted as “(d)” in the figure, but is mentioned as “(f)” in the description part.
7) in line 86, Scherrer equation was applied to analyse crystalline domain sizes, but it’s unclear for what is mentioned (crystal sizes of ZnO?).
8) in line 94, the sentence “the IR peaks of the solid samples are related to the vibration of ions in the photocatalysts structure” is queer to readers. If the authors believe this is valid, then please cite an appropriate reference therein. Otherwise, delete this sentence from the manuscript.
9) in line 95, please cite an appropriate reference for two peaks vibrational frequencies. Otherwise, this kind of discussion doesn’t make sense. In addition, be careful about the significant figures.
10) Similarly, in line 151, cite an appropriate reference for supporting “468 cm-1 due to the Zn sublattice and oxygen vibration of ZnO.”
11) in line 278, there is a description “UV light irradiation was applied by 500 w Xe lamb with 420 nm cut off filter,” but l > 420 nm means “visible light irradiation.” This discrepancy is confusing. In addition, what is lamb?
12) There are many typos, and sometimes the meaning is less clear. Before publication, ask for English correction. Some points are shown below:
In line 19: “hydrothermal” may be correctly “hydrothermally”
In line 43, lacking a parenthesis “)”
In line 49, it doesn’t make sense.
Author Response
Response to the Reviewers’ comments
Reviewer #2:
Major points:
Comments 1: “Magnetically” is not well supported: First, in line 242 the recycling using magnetic field is discussed. What’s happened without “magnetic field”? Discuss more details for this process, because this would be the most important part in this work. Second, in line 65, the authors mentioned to the magnetic properties in relation to the recyclability. Indeed, the photocatalytic durability was supported by data shown in Figure 7c. However, if the authors assert the magnetic property-related recyclability, it would be better for the authors to show the magnetic property of materials including saturation magnetization (ss) before and after the photochemical investigation. For example, ss of bulk magnetite is reported to be 90 emu g‒1 (Angew. Chem. Int. Ed. 2007, 46, 1222–1244), and its comparison and detailed experimental description would greatly support the “magnetic recyclability.”
Response: This is valuable advice for improving the quality of our manuscript. However, due to the lack of an instrumental facility, we couldn’t perform the VSM analysis at this time. We consider this a valuable suggestion and will do it for our future studies. Additionally, we gave the photographic images of magnetic separation of photocatalysts using an external magnetic field, as shown in Fig. 8d.
2) in line 173, the authors discussed the “improved absorption of visible light compared to the pristine.” However, this seems to be just due to the additional absorption by Fe3O4 (and “carbon”). Analysis for Figure 4b is also queer. In addition, ZnO itself also showed a certain photocatalytic performance as shown in Figure 6c. If the authors discuss the “seemingly narrowed” optical bandgap in relation to the improved photocatalysis, the authors must conduct photochemical investigations with, for example, l > 450 nm using a cut-filter. Otherwise, the whole discussion of the energy bandgap using Tack’s plot doesn’t make sense.
Response: I would like to thank the reviewer for this suggestion. Under light irradiation, ZnO exhibits photocatalytic performance, however, degradation performance is relatively low compared to other samples. MNC decreased the conduction and valance band positions, as well as the charge carrier recombination rate. As a result, the degradation performance of the prepared samples increases simultaneously. In addition, we did not use any cut-off filters. This was a manual error that has now been corrected.
3) In line 17, the authors mentioned “nanocomposites were designed and synthesized.” However, it was not clear for what is the design philosophy for this work (in other word, what is designed by how based on which idea?).
Response: Thanks for the valuable comments, we made a mistake with that sentence, and now we revise the manuscript as per the reviewer's suggestion.
4) The authors first mentioned to XRD and IR of materials, and then discussed the preparation. This sequence is less friendly for readers to understand the contents. Please first mention to how to prepare C, ZnO, and ZnO/MNC in the Results and Discussion part, with appropriate citations, if any.
Response: We thank to the reviewer for this suggestion and we have changed it in the revised manuscript.
5) “Advantage of carbon” is less clear in the introduction part: In line 63, the authors mentioned to “biomass derived carbon” and carbon aerogel for previous works, but the advantage and rational necessitation of using them is unclear to readers. And, what’s the advantage of this method over many other synthetic methods, including chemical vapor deposition (Chem. Sci. 2022, 13, 3140‒3146; Adv. Mater. 2022, 34, 2108750)? If the current method enjoys relatively cheap cost with relatively large surface area and enriched polar functional groups including hydroxyl and carbonyl groups as is discussed in lines 96-106, then they would also be great advantages of utilizing them in this work, and this should be mentioned in the introduction part for a “better reading material”.
Response: These is valuable comments to improve our the quality of the manuscript. We provide the advantages of carbon aerogel and synthetic methods. Further, the importance of the functional groups was discussed briefly in the introduction part and cited the below-mentioned reference in the revised manuscript.
[1]. Yamamoto, M.; Zhao, Q.; Goto, S.; Gu, Y.; Toriyama, T.; Yamamoto, T.; Nishihara, H.; Aziz, A.; Crespo-Otero, R.; Tommaso, D.D.; et al. Porous Nanographene Formation on γ-Alumina Nanoparticles via Transition-Metal-Free Methane Activation. Chemical Science 2022, 13, 3140–3146, doi:10.1039/D1SC06578E.
[2]. El-Hakam, S.A.; ALShorifi, F.T.; Salama, R.S.; Gamal, S.; El-Yazeed, W.S.A.; Ibrahim, A.A.; Ahmed, A.I. Application of Nanostructured Mesoporous Silica/ Bismuth Vanadate Composite Catalysts for the Degradation of Methylene Blue and Brilliant Green. Journal of Materials Research and Technology 2022, 18, 1963–1976, doi:10.1016/j.jmrt.2022.03.067.
6) In lines 74 & 77, there are description of Miller indices such as (100), (002), and (101). The parenthesis means a specific plane in definition. If the XRD patterns were obtained for isotropic powder, then the correct description will be 100, 002, and 101 rather than (100), (002), and (101). For details, please refer to a suitable text book. This would be the same for the selected area electron diffraction in Figure 3e, if further characterization is accomplished for the SAED.
Response: This is valuable and important suggestion to improve the quality of the manuscript. We are not provide the suitable reference, we acknowledge our sincere apology for this mistake. Foroughi et al., reported the similar XRD result of Template-Free Synthesis of ZnO/Fe3O4/Carbon Magnetic Nanocomposite: Nanotubes with Hexagonal Cross Sections and Their Electrocatalytic Property for Simultaneous Determination of Oxymorphone and Heroin. Similarly, our previous work reported the biomass derived carbon aerogel supported Fe3O4 nanocomposites for the removal of erichrome block-T.
[1]. Foroughi, M.M.; Jahani, S.; Aramesh-Boroujeni, Z.; Vakili Fathabadi, M.; Hashemipour Rafsanjani, H.; Rostaminasab Dolatabad, M. Template-Free Synthesis of ZnO/Fe3O4/Carbon Magnetic Nanocomposite: Nanotubes with Hexagonal Cross Sections and Their Electrocatalytic Property for Simultaneous Determination of Oxymorphone and Heroin. Microchemical Journal 2021, 170, 106679, doi:10.1016/j.microc.2021.106679.
[2]. Shanmugam, P.; Wei, W.; Qian, K.; Jiang, Z.; Lu, J.; Xie, J. Efficient Removal of Erichrome Black T with Biomass-Derived Magnetic Carbonaceous Aerogel Sponge. Materials Science and Engineering: B 2019, 248, 114387.
Minor points:
1) In Figure 3e, there are many spots in the selected area electron diffraction. Because the current work lacks EDS analysis for SEM/TEM, this SAED is a very nice source of identifying and characterizing the composites at nano-level. The referee recommends the authors to analyse the SAED for better characterization, if possible.
Response: Thank you for these valuable comments. As per the reviewer’s suggestion we have modified the SAED patterns discussions with more observed results.
2) In figure 5, what’s “Caron)”? If this is correctly carbon, then why do the carbon and magnetite show the photoluminescence similar to that of ZnO in the steady-state emission spectra? Maybe they are correctly Zn/MNC(##%), then unify the color in figures throughout the manuscript (For example, using the same colors for each material in Figure 4 and Figure 5 will help authors understand well the content).
Response: Please accept our deepest apologies for this error. In fact, we didn't take the UV and PL spectra for carbon nanoparticles, as we misrepresented the name in the figure. This manuscript has been corrected and the figures colors have also been changed in a similar way.
3) in line 55, the authors described “rapid electron recombination of photogenerated charge carrier,” but this may correctly be “rapid charge recombination of photogenerated hole-electron pair,” for example.
Response: Yes, we agree our mistakes, as per your suggestion, the sentence corrected in the revised manuscript.
4) in line 64 and more several times later, there are a word “CA,” but it was not defined throughout the manuscript.
Response: Thank you for your valuable comments, as per the reviewer comments the word CA was defined in the first use.
5) in lines 23 & 91 (and others), there is an unclear word like “ZnO/MNC(55%).” What does the percentage value mean? Clearly define and explain them at both the introduction and main text (otherwise readers cannot understand what the sentence means).
Response: Thank your valuable comments, the MNC abbreviation is magnetically recoverable nanocomposites. The composites containing 55% percentage of ZnO and 45 percentage of MNC.
6) A Raman spectrum in Figure 3 is denoted as “(d)” in the figure, but is mentioned as “(f)” in the description part.
Response: Yes, we agree our mistakes, thanks to the reviewer for this valuable suggestion, now we have changed it in the revised manuscript.
7) in line 86, Scherrer equation was applied to analyse crystalline domain sizes, but it’s unclear for what is mentioned (crystal sizes of ZnO?).
Response: Thank you for your valuable comments, we made a mistake. It's a crystallite size, not a crystal size now we have corrected it in the revised manuscript.
8) in line 94, the sentence “the IR peaks of the solid samples are related to the vibration of ions in the photocatalysts structure” is queer to readers. If the authors believe this is valid, then please cite an appropriate reference therein. Otherwise, delete this sentence from the manuscript.
Response: Yes, we agree our mistakes. We remove the sentence in the revised manuscript.
9) in line 95, please cite an appropriate reference for two peaks vibrational frequencies. Otherwise, this kind of discussion doesn’t make sense. In addition, be careful about the significant figures.
Response: Thank you for your valuable comments, we provide the appropriate reference for the two peaks. Additions, the clear FTIR spectra given in the revised manuscript.
[1]. Vijayalakshmi, D.; Chellappa, M.; Anjaneyulu, U.; Manivasagam, G.; Sethu, S. Influence of Coating Parameter and Sintering Atmosphere on the Corrosion Resistance Behavior of Electrophoretically Deposited Composite Coatings. Materials and Manufacturing Processes 2015, 31, doi:10.1080/10426914.2015.1070424.
[2]. Jayarambabu, N. Germination and Growth Characteristics of Mungbean Seeds (Vigna Radiata L.) Affected by Synthesized Zinc Oxide Nanoparticles. International Journal of Current Engineering and Technology 2014, 4, 5.
10) Similarly, in line 151, cite an appropriate reference for supporting “468 cm-1 due to the Zn sublattice and oxygen vibration of ZnO.”
Response: This is valuable suggestion to improve the quality of the manuscript, the appropriate reference and slightly modify the sentence in the revised manuscript.
[1]. Abdolhosseinzadeh, S.; Asgharzadeh, H.; Sadighikia, S.; Khataee, A. UV-Assisted Synthesis of Reduced Graphene Oxide–ZnO Nanorod Composites Immobilized on Zn Foil with Enhanced Photocatalytic Performance. Research on Chemical Intermediates 2016, doi:10.1007/s11164-015-2291-z.
11) in line 278, there is a description “UV light irradiation was applied by 500 w Xe lamb with 420 nm cut off filter,” but l > 420 nm means “visible light irradiation.” This discrepancy is confusing. In addition, what is lamb?
Response: We apologize for this mistake. Previously, we used halogen lamps with filters in some of our experiments. In this case, we are using a xenon lamp. In the revised version, we have corrected the manual error.
12) There are many typos, and sometimes the meaning is less clear. Before publication, ask for English correction. Some points are shown below:
Response: Thank you for your valuable comments, the English language and grammar was thoroughly checked native English speaker.
In line 19: “hydrothermal” may be correctly “hydrothermally”
Response: We thank to the reviewer for this valuable suggestion, we have changed in the revised manuscript.
In line 43, lacking a parenthesis “)”
Response: We thank to the reviewer for this valuable suggestion, suggestion We have changed it in the revised manuscript.
In line 49, it doesn’t make sense.
Response: As per the reviewer suggestion we have changed the sentence in the revised manuscript.
Reviewer 3 Report
This manuscript prepared hydrothermal-assisted magnetic ZnO/Carbon nanocomposites and used it in the photocatalytic degradation of methylene blue (MB) under Visible light irradiation. The as-synthesized composites were characterized by numerous techniques such as XRD, FTIR, SEM/EDAX, TEM, Raman and DRS-UV. The accomplished samples performance is impressive. Therefore, I would like to recommend published this work after addressing the following points:
1. The composite structure has good adsorption behavior performance, however, the relationship between properties and microstructure has not been discussed and analyzed in detail, thus, it makes the article seem shallow.
2. The Authors should also proofread their manuscript (some spelling and grammar errors).
3. - The conclusion is also not targeted to the important aspects described in the manuscript; please improve it.
4. The introduction section is not clear enough and it is difficult for the reader to find a relation of ideas with the text in general.
5. Authors are suggested to write a clear introduction and add more references where are missing and key publications on the supported nanoparticles as a photocatalysts: Some references could be cited such as: https://doi.org/10.1007/s10904-022-02389-8, https://doi.org/10.1016/j.heliyon.2022.e09652, https://doi.org/10.1021/acsomega.1c03735, https://doi.org/10.1016/j.jmrt.2022.03.067, https://doi.org/10.1016/j.jtice.2021.08.034.
6. Maybe the author should compare their results clearly with other reported works, highlighting the advantage and disadvantages of their novel composite.
7. The quality of the Figures and tables should be improved.
8. Caption in Figure (1) does not exist?
9. Caption in Fig. 3 that related to Raman analysis was wrong (f instead of d)?
10. Caption in Fig. 4, roman number or English letters????
Author Response
Response to the Reviewers’ comments
Reviewer III:
This manuscript prepared hydrothermal-assisted magnetic ZnO/Carbon nanocomposites and used it in the photocatalytic degradation of methylene blue (MB) under Visible light irradiation. The as-synthesized composites were characterized by numerous techniques such as XRD, FTIR, SEM/EDAX, TEM, Raman and DRS-UV. The accomplished samples performance is impressive. Therefore, I would like to recommend published this work after addressing the following points:
- The composite structure has good adsorption behavior performance, however, the relationship between properties and microstructure has not been discussed and analysed in detail, thus, it makes the article seem shallow.
Response: We thank to the reviewer for this valuable suggestion. In the SEM discussion part, we discussed the meso and micro porous structure of carbon materials in the revised manuscript.
- The Authors should also proofread their manuscript (some spelling and grammar errors).
Response: Thanks for your values suggestion, the throughout the manuscript corrected by native English speaker.
- The conclusion is also not targeted to the important aspects described in the manuscript; please improve it.
Response: as per the reviewers suggestion we have added more description in the conclusion part.
- The introduction section is not clear enough and it is difficult for the reader to find a relation of ideas with the text in general.
Response: As per the reviewer suggestion we have changed the introduction section in the revised manuscript.
- Authors are suggested to write a clear introduction and add more references where are missing and key publications on the supported nanoparticles as a photocatalysts: Some references could be cited such as: https://doi.org/10.1007/s10904-022-02389-8, https://doi.org/10.1016/j.heliyon.2022.e09652, https://doi.org/10.1021/acsomega.1c03735, https://doi.org/10.1016/j.jmrt.2022.03.067, https://doi.org/10.1016/j.jtice.2021.08.034.
Response: Thanks for your valuable comments to improve the quality of the manuscript. All the references are very interesting and improve the manuscript quality. As per the reviewer suggestion, we cited all the reference in appropriate position.
[1]. Alshorifi, F.; Ali, S.; Salama, R. Promotional Synergistic Effect of Cs–Au NPs on the Performance of Cs–Au/MgFe2O4 Catalysts in Catalysis 3,4-Dihydropyrimidin-2(1H)-Ones and Degradation of RhB Dye. Journal of Inorganic and Organometallic Polymers and Materials 2022, doi:10.1007/s10904-022-02389-8.
[2]. Alshorifi, F.T.; Alswat, A.A.; Salama, R.S. Gold-Selenide Quantum Dots Supported onto Cesium Ferrite Nanocomposites for the Efficient Degradation of Rhodamine B. Heliyon 2022, 8, e09652, doi:10.1016/j.heliyon.2022.e09652.
[3] Alshorifi, F.T.; Alswat, A.A.; Mannaa, M.A.; Alotaibi, M.T.; El-Bahy, S.M.; Salama, R.S. Facile and Green Synthesis of Silver Quantum Dots Immobilized onto a Polymeric CTS–PEO Blend for the Photocatalytic Degradation of p-Nitrophenol. ACS Omega 2021, 6, 30432–30441, doi:10.1021/acsomega.1c03735.
[4]. El-Hakam, S.A.; ALShorifi, F.T.; Salama, R.S.; Gamal, S.; El-Yazeed, W.S.A.; Ibrahim, A.A.; Ahmed, A.I. Application of Nanostructured Mesoporous Silica/ Bismuth Vanadate Composite Catalysts for the Degradation of Methylene Blue and Brilliant Green. Journal of Materials Research and Technology 2022, 18, 1963–1976, doi:10.1016/j.jmrt.2022.03.067.
[5]. Altass, H.M.; Morad, M.; Khder, A.E.-R.S.; Mannaa, M.A.; Jassas, R.S.; Alsimaree, A.A.; Ahmed, S.A.; Salama, Reda.S. Enhanced Catalytic Activity for CO Oxidation by Highly Active Pd Nanoparticles Supported on Reduced Graphene Oxide /Copper Metal Organic Framework. Journal of the Taiwan Institute of Chemical Engineers 2021, 128, 194–208, doi:10.1016/j.jtice.2021.08.034.
- Maybe the author should compare their results clearly with other reported works, highlighting the advantage and disadvantages of their novel composite.
Response: We thanks to the reviewer comments, we provide the separate table for the previous reports to comparative study.
- DÅ‚ugosz, O.; Szostak, K.; KrupiÅ„ski, M.; Banach, M. Synthesis of Fe3O4/ZnO Nanoparticles and Their Application for the Photodegradation of Anionic and Cationic Dyes. Int. J. Environ. Sci. Technol. 2021, 18, 561–574, doi:10.1007/s13762-020-02852-4.
- Elshypany, R.; Selim, H.; Zakaria, K.; Moustafa, A.H.; Sadeek, S.A.; Sharaa, S.I.; Raynaud, P.; Nada, A.A. Elaboration of Fe3O4/ZnO Nanocomposite with Highly Performance Photocatalytic Activity for Degradation Methylene Blue under Visible Light Irradiation. Environmental Technology & Innovation 2021, 23, 101710, doi:10.1016/j.eti.2021.101710.
- Lv, T.; Pan, L.; Liu, X.; Lu, T.; Zhu, G.; Sun, Z. Enhanced Photocatalytic Degradation of Methylene Blue by ZnO-Reduced Graphene Oxide Composite Synthesized via Microwave-Assisted Reaction. Journal of Alloys and Compounds 2011, 509, 10086–10091, doi:10.1016/j.jallcom.2011.08.045.
- Ngullie, R.C.; Alaswad, S.O.; Bhuvaneswari, K.; Shanmugam, P.; Pazhanivel, T.; Arunachalam, P. Synthesis and Characterization of Efficient ZnO/g-[C.Sub.3][N.Sub.4] Nanocomposites Photocatalyst for Photocatalytic Degradation of Methylene Blue. Coatings (Basel) 2020, 10.
- Photocatalytic Degradation of Methylene Blue Using Polymeric Membranes Based on Cellulose Acetate Impregnated with ZnO Nanostructures - PMC Available online: https://www.ncbi.nlm.nih.gov/pmc/articles/PMC8512553/ (accessed on 5 September 2022)
- The quality of the Figures and tables should be improved.
Response: As per the reviewer suggestions quality of the images has been changed
- Caption in Figure (1) does not exist?
Response: We thank to the reviewer for this suggestion and we have added it in the revised manuscript.
- Caption in Fig. 3 that related to Raman analysis was wrong (f instead of d)?
Response: We thank to the reviewer for this valuable suggestion now we have changed it in the revised manuscript.
- Caption in Fig. 4, roman number or English letters????
Response: We thank to the reviewer for this valuable suggestion now we have changed it in the revised manuscript.
Round 2
Reviewer 2 Report
The manuscript was improved after the first round of revisions, but there remains an important flaw to be improved:
In line 290, there is a description "After the first cycle, the photocatalysts were collected using external magnetic field and washed, dried, and recycled." The largest concern of Reviewer 2 is what's happened without external magnetic field.
That is, the current manuscript even lacks the description of "control experiments with no magnetic field." Description to show the advantage of applying the external magnetic field for sample collection/recovery is indispensable for this work to be published, because the title says "Magnetically Recoverable" which must be supported by experimental basis.
This argue was indeed described in the first review report by Reviewer 2, but the authors did not improve this, at least Reviewer 2 thinks. The followings are possible ways to deal with:
1) Describe detailed experimental procedures and outcome of "magnetically assisted photocatalyst collection." This includes description of control experiments lacking magnetic field, and its results.
2) For example, the authors can discuss like "With applied magnetic field, the recovery reached ##%, while the recovery of photocatalysts was much poor with ##% in the absence of external magnetic field, indicating the advantages and efficacy of applied magnetic field in this methodology and the photocatalyst design."
After addressing this, the manuscript shall be accepted by Catalysts.
Author Response
To 10.09.20222
Prof. Angela Xue
Section Managing Editor,
Catalysts-MDPI
Manuscript Ref. No.: catalysts-1896015
Title: Magnetically Recoverable Biomass Carbon Aerogel Supported ZnO
(ZnO/MNC) Composites for the Photodegradation of Methylene Blue.
Dear Editor,
Thank you for your email concerning the above-mentioned manuscript. We thank you and the Reviewers for valuable comments to improve the quality of our manuscript. We have revised the manuscript addressing the Reviewers’ comments. We have highlighted the changes in the revised manuscript. We trust that the revised manuscript is acceptable for publication in Catalysts. Below, we have provided our responses to the reviewers’ comments.
Sincerely,
Dr. P. Shanmugam M.Sc., Ph.D
Post Doctoral Researcher
Department of Chemistry
Faculty of Science
Thammasat University
Pathum thani-12120, Thailand

Reviewer 3 Report
The manuscript is acceptable in the present form since authors have well addressed the questions proposed by referees.
Author Response
Dear Reviewer
Greetings!
Thanks for your valuable comments and accepted our manuscript.
Thank you
Dr. P. Shanmugam